# Brain Connectivity Analysis Under Semantic Vigilance and Enhanced Mental States

**DOI:** 10.3390/brainsci9120363

**Published:** 2019-12-09

**Authors:** Fares Al-Shargie, Usman Tariq, Omnia Hassanin, Hasan Mir, Fabio Babiloni, Hasan Al-Nashash

**Affiliations:** 1Biosciences and Bioengineering Research Institute, Department of Electrical Engineering, American University of Sharjah, P.O. Box 26666, Sharjah, UAE; utariq@aus.edu (U.T.); g00060182@aus.edu (O.H.); hmir@aus.edu (H.M.); hnashash@aus.edu (H.A.-N.); 2Department Molecular Medicine, University of Sapienza Rome, 00185 Rome, Italy; fabio.babiloni@uniroma1.it; 3College Computer Science and Technology, University Hangzhou Dianzi, Hangzhou 310018, China

**Keywords:** vigilance, enhancement, EEG, partial directed coherence (PDC), graph theory analysis (GTA)

## Abstract

In this paper, we present a method to quantify the coupling between brain regions under vigilance and enhanced mental states by utilizing partial directed coherence (PDC) and graph theory analysis (GTA). The vigilance state is induced using a modified version of stroop color-word task (SCWT) while the enhancement state is based on audio stimulation with a pure tone of 250 Hz. The audio stimulation was presented to the right and left ears simultaneously for one-hour while participants perform the SCWT. The quantification of mental states was performed by means of statistical analysis of indexes based on GTA, behavioral responses of time-on-task (TOT), and Brunel Mood Scale (BRMUS). The results show that PDC is very sensitive to vigilance decrement and shows that the brain connectivity network is significantly reduced with increasing TOT, *p <* 0.05. Meanwhile, during the enhanced state, the connectivity network maintains high connectivity as time passes and shows significant improvements compared to vigilance state. The audio stimulation enhances the connectivity network over the frontal and parietal regions and the right hemisphere. The increase in the connectivity network correlates with individual differences in the magnitude of the vigilance enhancement assessed by response time to stimuli. Our results provide evidence for enhancement of cognitive processing efficiency with audio stimulation. The BRMUS was used to evaluate the emotional states of vigilance task before and after using the audio stimulation. BRMUS factors, such as fatigue, depression, and anger, significantly decrease in the enhancement group compared to vigilance group. On the other hand, happy and calmness factors increased with audio stimulation, *p <* 0.05.

## 1. Introduction

Vigilance can be defined as the mental capacity to sustain attention over an extended period. We define a vigilance task as a cognitive task in which a subject is supposed to stay vigilant. A typical finding in vigilance tasks is the phenomenon of vigilance decrement. This refers to the decline in target detection rate or speed of response over time-on-task (TOT). Retaining attention above a constant level is of vital importance in many applications. Some examples of entities that need sustained vigilance include pilots in aircraft [1], drivers operating vehicles [2], security forces in defense systems [3], cyber intrusion detection [4], industrial process control settings [5], and real classroom settings [6]. In all these applications, individuals are involved in repetitive, monotonous, and long-term high demanding cognitive tasks, leading to lowered productivity and increased safety risks. Hence, an effective method of detecting vigilance decrement is needed to prevent vigilance-related risks. This is indirectly linked to enhance quality of life as well. To understand the vigilance decrement, studies have presented three different theories: overload, underload, and motivational control theories [7,8,9]. Overload theory claims that vigilance tasks are extremely demanding so much so that the limited neurocognitive resources required for maintaining vigilance are exhausted more rapidly than they can be replenished, causing vigilance decrement. In contrast, underload theory argues that the lack of stimulation or the monotony of vigilance tasks causes attention to shift from the external task and leads to increase boredom, task disengagement, and poor performance. Meanwhile, motivational control theory links task performance with the expected value of outcomes; that is, a TOT related behavioral decline would be expected if the performance becomes more costly than the expected value.

Previous studies have reported several factors affecting the timing and magnitude of the vigilance decrement. These include signal duration, source complexity, and use of declarative memory in the task [10,11]. Stimulus event rate has also been shown to impact the amount of decrement in modified vigilance tasks, e.g., in an n-back working memory-updating task [12]. In fact, the timing of vigilance decrement varies depending on the task demand level. Performance decline typically occurs within the first 20–35 min of the task, with 50% of the decrement observed in the first 15 min [13,14]. The onset of the decline may come sooner in more challenging tasks [15,16]. Despite the assortment of factors that influence the vigilance decrement, theoretical debates about the underlying mechanisms have not been resolved. Researchers have utilized physiological measures to elucidate debates regarding the vigilance decrement with a variety of imaging techniques [17,18,19,20,21,22]. Previous neuroimaging studies on vigilance have uncovered a network of regions associated with arousal and attention, including a top-down right-lateralized network of fronto-parietal regions and areas providing bottom-up input including the thalamus and basal forebrain [23,24]. As TOT increases, cerebral blood flow tends to decrease. This possibly reflects a depletion of neural resources or an inability to retrieve these resources. Many methods for vigilance detection have been proposed in the literature [14,17,18,19,20,21,22,25]. Converging evidence using behavioral, neural, and subjective measures shows that vigilance requires hard mental work and is stressful [15]. In general, the existing approaches do not provide any solution to measure vigilance level quantitatively. Identifying reliable and valid biomarkers remains a challenge within the research community. Among numerous physiological indicators, Electroencephalography (EEG) signals may be one of the most predictive indicators of state discrimination because all the physical and mental activities associated with vigilance are reflected in EEG.

EEG is the most popularly used technique for vigilance estimation due to its portability, high temporal resolution, low-cost properties, and low constraints on participants’ behavioral performance. The EEG oscillations classified by their frequencies have the following approximate ranges: (1–4) Hz for delta, (4–8) Hz for theta, (8–13) Hz for alpha, (13–30) Hz for beta and higher than 30 Hz for gamma. It has been suggested that these frequency ranges could be individually determined by a precise procedure involving the detection of the alpha peak, also known as Individual Alpha Frequency [26]. Generally, it has been shown that increasing TOT leads to observable changes in both ongoing EEG activity and event-related potential (ERP) locked to the task stimuli. In ERP, when a person enters a state of mental fatigue, the amplitude of the ERP components associated with error monitoring and inhibition is significantly reduced [27,28,29,30,31,32]. In most of the ERP studies, consistent finding is that the magnitude of P300 components decreased with vigilance decrement. Correspondingly, recent studies have reported a correlation between EEG rhythmic components and vigilance levels [33,34,35,36]. For example, studies have shown that activities in the alpha rhythm increased over the frontal and parietal-occipital areas with increasing TOT [37,38,39]. The increase in alpha rhythm was also associated with decreased in concentration and alertness. Consequently, Martel and his colleagues [37] observed an increased activity in the alpha frequency range (8–14 Hz) for vigilance decrement. They proposed predicting attention lapses in a convert setting up to 10 s in advance. Progressive increases near 4 Hz theta and decreases near 40 Hz gamma have specifically been correlated with decreasing arousal and alertness during periods when the attentional system is challenged [40]. However, most research in the vigilance paradigm has utilized sensory-based tasks, whereas little work has focused on the effects in relation to a cognitive-based task or tasks that involve both cognitive and sensory processes such as semantic vigilance tasks [14,41]. Besides, these methods only describe the EEG signals of a single channel in a local brain region and do not involve the functional connection between brain regions.

The EEG signal contains information from a complex and dense network of billions of interconnected neurons. In order to capture the inter-regional interactions, recent studies have utilized functional connectivity with graph theory analysis (GTA) metrics to assess driving drowsiness and mental fatigue [42,43]. In the context of GTA, brain networks can be interpreted as a graph, with different anatomical and/or functional brain regions represented by nodes and any interaction represented by links between each pair of brain regions. One fatigue study has reported weakened nonlinear interdependence between each electrode in theta-range, alpha-range and beta-range at the end of a driving epoch [44]. Meanwhile, another study found that when mental fatigue levels increase, the parietal-to-frontal functional coupling of the alpha frequency band is weakened, and the frontal-to-central functional coupling of the beta frequency band is heightened in left brain area [45]. It is not clear whether vigilance affects the connectivity network at specific frequency band or affects the entire spectrum. In this paper, we hypothesize that performing semantic vigilance task weakens connectivity network of full spectrum at range (0.5–30) Hz as time passes. Semantic vigilance tasks require participants to attend to and process characters, symbols, text-speak words, words, or non-words over extended periods of time [46,47]. The semantic vigilance tasks require operators to respond to targets that are semantic or lexical in nature and withhold response to neutral stimuli, which are not semantically representative or related to target signals. Thus, semantic vigilance tasks are unique in that they do not fall neatly into the cognitive sensory vigilance distinction and could arguably involve both cognitive and sensory processes given the nature and design of the vigilance task. Thus, the central theme of this research work is to quantify the level of vigilance decrement and to provide a solution to minimize vigilance decrement using audio stimulation. This research proposes a novel protocol to induce semantic vigilance using modified Stroop color word test (SCWT) and minimize vigilance decrement using an auditory stimulation of pure tone at 250 Hz. The pure tone is an auditory illusion perceived by the brain when one 250-Hz frequency of sound is played into the right and left ears simultaneously. We expect the pure tone stimulation can influence behavior and cognition in a variety of ways.

## 2. Materials and Methods

This section describes the participants’ information, stimuli and data acquisition process. It also describes the pre-processing of the acquired EEG data, audio stimulation, method used to estimate functional connectivity, application of graph theory, and statistical analysis.

### 2.1. Participants

Participants were 24 healthy young students from the American University of Sharjah (17-male and 7-female, age: 22 ± 2 years, (mean ± standard deviation)). All participants had normal or corrected to normal vision and no reported hearing deficits/difficulties. In addition, they had no history of neurological or psychiatric illnesses and had no current or prior psychoactive medication use. The experiment was conducted between 3.00 pm and 7 pm to avoid the influences of circadian rhythm on cognitive vigilance performance [48]. The aims and procedures of the experiment were explained to all the participants before commencing the experiment. All participants were asked to give written informed consent before participation in the study. The participants were free to stop their participation during the experiment or to withdraw from the experiment for any reason. All participants were asked to abstain from caffeine, exercise, energy drink, and tobacco use for 24 h before testing. All methods performed follow the Declaration of Helsinki. The protocol of the study was approved by the Institutional Review Board (IRB) committee at the American University of Sharjah.

### 2.2. Experiment Design

#### 2.2.1. Semantic Vigilance Task

The vigilance task used in this study is based on the computerized version of the Stroop color-word test (SCWT), developed in our laboratory at American University of Sharjah, Sharjah, UAE. The task is designed and presented in MATLAB with a Graphical User Interface (GUI). It consists of displaying six color words such as (‘Blue’, ‘Green’, ‘Red’, ‘Magenta’, ‘Cyan’, and ‘Yellow’) in random order. Only one word is displayed at a time and the answers of the color word to be matched to are presented in random sequences in the computer screen monitor. The displayed color word on the monitor screen is written in a different color than the word’s meaning, and the correct answer is the color in which the word is displayed (e.g., if Green is written in Cyan, then Cyan is the correct answer). The participants picked their answers as quickly and accurately as possible by left-clicking the mouse on one of the six answering buttons as shown in Figure 1a. The answer to the color word is presented with random colored background to increase the attention level to the task. Additionally, the colored background may mimic real-life challenges, e.g., finding an object in a difficult background. This may lead to reach vigilance decrement in a short time. Besides, participants were exposed to a mock user performance indicator that implied a poor performance by the participants in comparison with their peers, as shown in Figure 1 (in green and red horizontal bars). Furthermore, answering incorrectly or failing to answer each question within the time limit, the participants would receive feedback, i.e., a message of “Correct’’ or “Incorrect” or “Time is out” is displayed on the monitor. The aim for such a feedback was to add more visual vigilance in the participants.

Behavioral data such as reaction time to stimuli and accuracy of detection were collected while solving the task. This was then used to objectively measure vigilance decrement/enhancement. In this task, four indicators measure participants’ attention levels: commission error, omission error, reaction time, and accuracy. A commission error occurs when a participant fails to inhibit the response and incorrectly responds to a non-color word, whereas an omission error occurs when a participant is unable to pick-up or respond to the color word. Once participants’ responses are checked, the time they spent is recorded. The reaction time is measured as the average time it takes for the participant to respond correctly to a target stimulus. The number of trials also depended on the participant’s rating speed. Different markers were sent to mark the start and the end of epochs in each SCWT question. The overall accuracy was calculated based on the number of the color word correctly matched over the total number of the displayed color word targets. Note that the commission error is a measure of the level of impulsiveness, omission error reflects the level of inattention, and the reaction time value represents the speed of reaction.

The experimental protocol was conducted in two different scenarios to the ‘vigilance’ group and to the ‘enhancement’ group. In the vigilance group, each participant performs the SCWT continuously for 60 min, whereas, in the enhancement group, each participant performs the same SCWT while listening to an audio stimulation of pure tone (PT) at 250 Hz for 60 min. Typically, the frequency of 250 Hz is a good compromise between sensitivity and pleasing sounds.

The overall experimental time for each participant included 6 min for training and filling questionnaires, 2 min pre-baseline, 60 min with SCWT without audio/250 Hz, 2 min post-baseline and 5 min filling another questionnaire. Figure 1b shows the time window of the experiment.

#### 2.2.2. Audio Stimulation

Matlab software (R2018a, Natick, MA, USA) is used to produce the pure tone of 250 Hz. The tone is presented to the right and left ears of the participants using stereo headphones (MDR-NC7, Sony, Sharjah, United Arab Emirates). At the commencement of the experiment, the volume of the auditory stimuli is set by the participants. The stimuli are delivered at minimum intensities of 50 dB above threshold. The tone is presented continuously (no pulses) to avoid the impact of short rest on vigilance level [14]. Stimuli are generated at a sampling rate of 48 kHz to ensure that the highest stimulus frequencies are well below the Nyquist rate. Stimuli are represented in memory as 32-bit integers so that the full dynamic range could be tested without risk of quantization distortion.

### 2.3. Subjective Evaluation

All participants were asked to fill-in a questionnaire about their emotional that used a Brunel Mood Scale (BRMUS) [49]. The participants filled-in the questionnaires before they performed the semantic vigilance task and after they finished the experiment. The BRMUS is a self-report emotional state questionnaire composed of 32 items commonly used in literature for assessing the emotional state of the participants. These items correspond to an 8-factor model including “Anger,” “Tension”, “Confusion,” “Depression,” “Fatigue,” “Happy,” “Calmness”, and “Vigor.” Each item has 5-point Likert scale ranges from 0 to 4 representing “not at all” to “extremely” depending on the participant’s feelings.

### 2.4. EEG Data Acquistion and Pre-Processing

Electrophysiological data were recorded using 64 Ag/AgCl scalp electrodes arranged according to the standard 10–20 system (with ANT Neuro waveguard system and ASA Lab 4.9.2 acquisition software, ANT Neuro, Hengelo, Netherlands). The EEG data were acquired at a sampling rate of 500 Hz. The impedances of all EEG electrodes were maintained below 10 kΩ and referenced to the left and right mastoids, M1 and M2. Figure 2 shows a picture of the data acquisition and experimental set-up.

The recorded EEG signals were preprocessed using EEGLAB toolboxes (9.0.4) at Sharjah, United Arab Emirates [50] as well as using custom script [51,52]. The EEG signals were re-referenced to the common average reference and segmented into target-related EEG epochs of 1200 ms. Baseline removal and DC offset were performed by subtracting the mean from the data. All EEG signals were band-pass filtered using a finite impulse response (FIR) filter with 0.1 Hz to 30 Hz bandwidth. The abnormal epochs were manually removed. The eye-blinks and eye-movements were removed by utilizing independent component analysis (ICA) Infomax algorithm. The components representing artifacts, such as eye blinks, eye movements, and muscular activities were removed, and the remaining components were used to reconstruct the clean EEG signals. Typically, only one or two independent components relevant to eye blinks or eye movements were removed for each subject. Finally, all EEG epochs were visually double checked to ensure the quality of the EEG data. Further analysis would only include the correctly performed trials that were artifact-free in all channels. We binned epochs into three blocks, each of 20 min, to assess changes in connectivity network with time-on-task. A time block from 0–20 min is assigned to vigilance/Enhancement Level 1 (V1/E1), a time block from 21–40 min is assigned to Vigilance/Enhancement Level 2 (V2/E2), and a time-on-task from 41–60 min is assigned for Vigilance/Enhancement Level 3 (V3/E3), respectively. Then we investigated the connectivity network analysis for each time block separately.

### 2.5. Functional Connectivity

Partial directed coherence (PDC) is a multivariate autoregressive model (MVAR) used to estimate the directed flows of information among multichannel EEG signals. A multivariate model with m channels of EEG signals and order *p* is defined by Equation (1):(1)$$X(t)=\sum_{r=1}^{p}A(r)x(t-r)+E(t),$$
where *x(t)* is the vector of *m* channels of EEG signals at time *t* and the matrix *A(r)* contains the *r^th^* order autoregressive parameters. The variable ***E****(t)* represents the estimated error, which is supposed to be an uncorrelated Gaussian process with zero mean. Akaike information criterion (AIC) [53] is used to determine the proper model order *p* for our study; we used *p = 7.* Once the coefficients of the MVAR model are adequately estimated, we then obtain the ***A****(f)* using Equation (2):(2)$$A(f)=\sum_{r=1}^{p}A(r)e^{-i2\pi fr},$$

Finally, the PDC from channel *j* to *i* is given by Equation (3):(3)$$PDC_{i,j}(f)=\frac{{\bar{A}}_{i,j}(f)}{\sqrt{{\bar{a}}_{j}^{H}(f){\bar{a}}_{j}(f)}},$$
where Ā(*f*) is the transfer function, ā*_i_*(*f*) (*i* = 1, 2, …*M*) is the *i*^th^ column of the matrix Ā(*f*), *H* is the complex conjugate operation, and *PDC_i,j_* indicates the direction and weight of the information flow from channel *j* to *i* at the frequency *f*. Given that the sliding time window techniques might disclose higher resolution dynamics, a sliding time window of 1200 ms of EEG data is used in this study. Finally, one PDC network (62 × 62 weighted directed adjacency matrix) is created for each window in the total (0.5–30 Hz) frequency band. The PDC ranges in the interval (0, 1), with higher values indicating higher interaction between the two nodes. The process was applied to all subjects, and the grand average was then reported under different thresholding values.

### 2.6. Graph Theory Analysis

Graph theoretical analysis is aimed at providing a wide variety of quantitative measurements for assessing the topological architecture of a network. For a defined network or graph G, there are N nodes and W weighted edges; the global and local network metrics are the output of the graph theory analysis. In our case, the nodes are electrodes and links established by the connectivity measure of PDC. The brain network parameters are measured from the matrices obtained by PDC for each subject. In this study, the graphical analysis measures are based on the concepts of node degree, node strength, clustering coefficient, and characteristic path length and were calculated using open software toolboxes [54,55]. In particular, we infer a set of weighted directed networks over sparsity ranging from 10% to 30% with an interval of 0.05 and then calculated the integrals of these measures within the sparsity range.

#### 2.6.1. Nodal Degree ($d_{i}$)

This is the number of edges linked directly to a particular node and estimated using Equation (4) where the elements $a_{ij}$ refer to the edge status between node i and node j (i.e., $a_{ij}=1$ if $w_{ij}\neq 0$). A node i existing within a directed graph can have an indegree and an outdegree corresponding to the number of incoming and outcoming edges, respectively. The total degree of a node is simply the sum of the in- and out- degrees. For a brain functional network, degree centrality (measured by nodal degree) reflects the cerebral cortex regions that play a key role in the information transmission and processing of the brain [32]. If the degree centrality of a node is large, it means that the node has more connected nodes in the topology of the brain functional network. This indicates that it has an important position in the network. Therefore, the degree centrality can quantitatively analyze the importance of the node in the brain functional network.

(4)$$d_{i}{}^{tot}=d_{i}^{in}+d_{i}^{out}=\sum_{j\neq i\in N}a_{ji}+\sum_{j\neq i\in N}a_{ij},$$

#### 2.6.2. Nodal Strength ($NS_{i}$)

Nodal strength is the sum of edge weights that link a node to the other nodes in a weighted network and measured using Equation (5). The average node strength of the network is the overall arithmetic average of the strengths of all the individual nodes in the network. It represents the sum of all incoming and outgoing edge weights. In this case, node strengths help to find the involvement and information flow of a particular region in a functional brain network.

(5)$$NS_{i}=\sum_{j\in G}w_{ij}+\sum_{j\in G}w_{ji},$$
where $w_{ji}$ is an element weight of the PDC matrix.

#### 2.6.3. Clustering Coefficient (CC)

It is a measure of network segregation which quantifies the tendency to which neighboring nodes form complete networks or cliques. For a node i, the local clustering coefficient *C_i_* is calculated as the ratio between the sum of geometric means of all existing weighted triangles and the number of all possible triangles [54,56]. This relation can be mathematically expressed as in Equation (6). In this context, *C_i_* measures how well the cluster of node communicates, and a high value of *C_i_* relates to the high local efficiency of information transfer. The global clustering coefficient CC of a network is the arithmetic average of local coefficients of all nodes. It is calculated using Equation (7).

(6)$$C_{i}=\frac{\sum_{j\neq i\in N}\sum_{h\neq i\in N,h\neq j\in N}\left(w_{ij}{}^{1/3}+w_{ji}{}^{1/3})(w_{ih}{}^{1/3}+w_{hi}{}^{1/3})(w_{jh}{}^{1/3}+w_{hj}{}^{1/3}\right)}{d_{i}{}^{tot}(d_{i}{}^{tot}-1)-2\sum_{i\neq j}a_{ji}a_{ij}},$$

(7)$$CC=\sum_{i\in N}C_{i},$$

#### 2.6.4. Characteristics Path Length (PL)

It is a measure of graph integration and is calculated based on the minimum number of directed edges that must be traversed to go from one node to another [54]. The characteristic path of length L of a graph can be calculated as the average of the shortest paths between all pairs of nodes. It is defined by Equation (8).

(8)$$PL=\frac{1}{N}\sum_{j\neq i\in N}L_{ij},$$
where $L_{ij}$ is the minimum length of edges between node i and j. The edge lengths can be obtained as the reciprocal of edge weights. Note that the PL indicates the high global efficiency of information transfer.

#### 2.6.5. Hemispheric Information Flow (HIF)

This is performed to identify the information flow patterns within and between the left and right hemispheres. In this analysis, we only considered the lateralized electrodes/nodes at the left hemisphere (LH) and the one at the right hemisphere (RH). Electrodes/nodes located at the central/middle (Fpz, Fz, Fcz, Cz, CPz, Pz, POz, and Oz) were neglected. There was a total of 54 electrodes; 27 electrodes belong to the right hemisphere and 27 electrodes belong to the left hemisphere. Four sub-network connectivity matrices of size 27 were generated to represent the information flow between electrodes in LH to LH, LH to RH, RH to RH, and RH to LH. The total information flow from one node to all other nodes was calculated by row-wise summation of each sub-network connectivity matrices.

### 2.7. Statistical Analysis

Statistical analysis was performed in-between vigilance levels and between vigilance and enhancement at each level of time-on-task, independently. We performed pairwise t-tests (we used Bonferroni–Holm technique for correction [57]) to compare the connectivity metrics between vigilance and enhancement using the following criteria. First, we compared the connectivity metrics in nodal degree (d_i_), nodal strength (NS_i_), clustering coefficient (CC), path length (PL), and hemispheric information flow (HIF) between vigilance level 1 versus vigilance level 2 (V1 vs. V2); vigilance level 1 versus vigilance level 3 (V1 vs. V3); and vigilance level 2 versus vigilance level 3 (V2 vs. V3), respectively. Second, we compared the connectivity metrics in N_i_, NS_i_, CC, PL, and HIF between vigilance and enhancement at the three levels of time-on-task (V1 vs. E1; V2 vs. E2; and V3 vs. E3). A criterion of *p* < 0.05 was selected.

## 3. Results

### 3.1. Behavioral Data

To evaluate the performance of participants, the reaction time in response to stimuli of the task, accuracy of detection, omission error, and commission error are used. To do so, we divided data from the 60-min experiment into twenty 3 min blocks. In line with past vigilance studies, we examined how short is the response time, and accuracy varied with time-on-task. The results of the aforementioned variables are presented and discussed below.

#### 3.1.1. Reaction Time

The average reaction time increased with time-on-task during the vigilance phase and decreased with time-on-task during the enhancement phase as shown in Figure 3a. The linear slope of the reciprocal of reaction times showed that there was a decline in performance on average over the 60 min period of vigilance, suggesting a clear time-on-task effect with steadily increased reaction time over the course of the experiment. Meanwhile, in the enhancement phase, the linear slope of the reciprocal of reaction times showed an improvement over time. The statistical analysis showed that the speed of answering the task improved over time in the enhancement phase compared to the vigilance phase, *p* < 0.05. On average, pure tone stimulation at 250 Hz showed 21.82% improvement on participant’s performance over time.

#### 3.1.2. Accuracy

The analysis revealed that percentage of accuracy declined over time during the vigilance phase and increased or maintained high during the enhancement phase as shown in Figure 3b. The statistical analysis showed that the overall detection rate significantly improved with *p* < 0.01. On average, enhancement with pure tone stimulation showed 13.74% improvement on participant’s detection rate/accuracy.

#### 3.1.3. Omission Error

The percentage of omission errors were computed for each participant, and the average of all participants with the standard deviation is shown in Figure 4. The results of the omission errors showed decrement from vigilance phase (blue color) to enhancement phase (green color). The *t*-test comparisons revealed that omission error significantly decreased during the enhancement phase compared to vigilance phase, *p* < 0.0027.

#### 3.1.4. Commission Error

The percentage of commission errors were computed for each participant, and the average of all participants with the standard deviation is shown in Figure 4. The result of the commission error showed significant decrease from vigilance phase to enhancement phase, *p* < 0.05.

### 3.2. Subjective Data

The emotional states of the participants in the vigilance group/phase and in the enhancement group changed after the experimental procedures had finished compared to that of baselines. The statistical analysis showed that anger, tension, depression, vigor, fatigue, and confusion have significantly increased *p <* 0.05, while happy, and calmness were significantly decreased compared to baseline, *p <* 0.5.

Meanwhile, the statistical analysis between the vigilance group and the enhancement group showed that anger, depression, confusion, and fatigue, were significantly decreased *p <* 0.05, while happy and calmness were significantly increased from vigilance to enhancement, *p <* 0.01. The rest of the factors, including tension and vigor, were not significantly different. Table 1 summarizes the overall BRMUS mean evaluation scores after vigilance and after enhancement with their significant differences; increase (up arrows) and decrease (down arrows) level.

### 3.3. EEG Connectivity

The weighted directed connectivity network showed decrement (weakened) with increasing time-on-task in the vigilance group and increment (strengthen) or maintaining high in the enhancement group. The average weighted directed connectivity network patterns at 0.2 threshold measured from the PDC matrices are shown in Figure 5. Figure 5a shows the average directed connectivity network of vigilance group while Figure 5b shows the average directed connectivity network of the enhancement group at three levels of time-on-task: Level 1 (0–20 min), Level 2 (21–40 min), and Level 3 (41–60 min). Interestingly, the network pattern shows decrement in the degree of connectivity as well as in the weight with time-on-task in the vigilance group. In addition, large quantities of edges between parietal and temporal regions gradually vanish (this is supported by the statistical analysis of the nodal degree in Figure 6 and the nodal strength in Figure 7). In particular, electrodes namely PT8, T8, P8, and P6 show the highest sensitivity to the task (based on the layout of the 64 electrodes in the 10–20 system). It can be noted that, with increasing time-on-task, the connected edges are relatively concentrated in left-brain areas in the first 40 min and shift the processing to a more bilateral style in the last 20 min. Meanwhile, in the enhancement group, the network maintains high connectivity between the nodes at the three levels of time-on-task. The results show similar complex network, linked by many edges, between frontal, temporal, central, parietal, and occipital regions. The higher the value of the weight in the network, the more the information flows from one node to another.

### 3.4. Graph Theory Analysis of Brain Network

#### 3.4.1. Nodal Degree Index

The nodal degree index was computed for each electrode and each subject at six different thresholding values (at 0.0, 0.1; 0.15, 0.20, 0.25, and 0.30) and then averaged for all subjects and electrodes. Figure 8a shows the normalized average nodal degree index under three levels of vigilance at different thresholding values. It can be noted that the degree is almost unchanged when thresholding values are very small, e.g., at 0.0, while the degree values decrease when the thresholding values increase. This could be caused by the gradual disappearance of connected edges. In addition, it is clear that the network structure is sensitive to the threshold. Altogether, the average nodal degree index of the network shows a significant decrease with increasing the time-on-task from vigilance Level 1 to vigilance Level 2 to vigilance Level 3 at thresholding values of 0.10, 0.15, 0.20, 0.25, and 0.30, *p* < 0.05. On the contrary, Figure 8b shows the average nodal degree index under enhancement at different thresholding values. Interestingly, the average nodal degree in the enhancement group was higher than that in the vigilance group. This indicates that the audio stimulation enhanced the connectivity network of the brain. We also performed statistical analysis between the nodal degree indexes, in-between vigilance levels, between vigilance and enhancement groups. The statistical analysis shows significant increase in the nodal degree from vigilance to enhancement at the three levels of time-on-task (at thresholding values between 0.10 to 0.30) with mean *p*-value of less than 0.05, *p* < 0.05 demonstrated by the * sign. To further validate that, we performed another statistical analysis between vigilance levels and vigilance and enhancement based on local node degree. Figure 6b shows the T-maps between vigilance levels based on local node degree and Figure 6b shows the T-maps between vigilance and enhancement at the three levels of TOT. The average normalized node degrees in the three levels of vigilance located on the left hemisphere are higher than that in the right hemisphere as shown in Figure 9. The statistical analysis also supports our claim that right hemisphere is sensitive to vigilance decrements and the differences between left and right hemisphere in the first two levels of TOT are significant as demonstrated by the ** sign.

In addition, the overall statistical analysis between vigilance decrement group and enhancement group (by taking all the levels together) based on nod degree shows that there was a significant difference between mental states as shown in Figure 10. This indicates that the audio stimulation significantly enhances the mental state regardless the levels of TOT. This also supports our primary claim that audio can reduce vigilance decrement. From Figure 10, it is very clear that the right hemisphere is highly sensitive to vigilance decrement. In this context, the group analysis supports the results obtained by individual level analysis (the analysis between the three levels of TOT).

#### 3.4.2. Nodal Strength

Figure 11 gives the average nodal strengths values (mean ± std across all subjects) of the vigilance group for all electrodes at the three levels of time-on-task. Figure 11a shows node strength at vigilance level 1 at time-on-task between 0–20 min. From Figure 11, we can observe that vigilance group has more node strength within the left hemisphere compared to the right hemisphere. Similarly, Figure 11b,c shows the nodal strength at vigilance level 2 and level 3, respectively. Looking at the 3 levels of vigilance with time-on-task, it is clear that the nodal strengths decrease from level 1 to level 2 and from level 2 to level 3. The decrease in the nodal strengths indicates that the information flow decreases with increasing the time-on-task or with increasing the vigilance levels. In addition, lower values of particular nodal strengths indicate that the node sends and receives less information.

Similarly, Figure 12 gives the average values of nodal strengths (mean ± std across all subjects) of the enhancement group for all electrodes. Figure 12a shows the average nodal strength of enhancement group at level 1 of time-on-task between 0–20 min. The information flow is high in both, right and left hemisphere, with slight increase in the left hemisphere over the frontal cortex. The higher nodal strength in the frontal regions indicates the use of short-term memory [58]. Similarly, Figure 12b,c shows the nodal strengths of enhancement group at level 2 and level 3 with time-on-task between 21–40 min and 41–60 min, respectively. The enhancement group’s node strength is maintained approximately the same with time-on-task at level 1, level 2, and level 3, which means that the brain is highly active and the information flow/transfer remains high. The higher values of the nodal strengths indicate that each node sends and receives more information.

Figure 7a shows the statistical analysis between vigilance levels based on the normalized node strength represented by their t-maps. The T-map in Figure 7 is obtained based on the differences between normalized node strengths of vigilance levels (V1 vs. V2; V1 vs. V3; and V2 vs. V3) and vigilance and enhancement (V1 vs. E1; V2 vs. E2; and V3 vs. E3). A t-value of to ≥ 2.5 (corresponding to *p* < 0.05) indicates that the differences were significant. The topographical maps demonstrated that the right hemisphere was the most sensitive brain region to vigilance decrement, and its connectivity network decreased with time-on-task as shown earlier in Figure 5a. Similarly, Figure 7b shows the statistical analysis (T-maps) between the vigilance and enhancement groups at the three levels of time-on-task. The statistical analysis in Figure 7 shows that the audio stimulation significantly enhances the flow of information, specifically over the frontal and parietal regions and the right hemisphere as demonstrated by the T-maps. The higher T-values in right hemisphere in Figure 7a indicate that it was highly sensitive to vigilance decrements. This statistical analysis is also in line with the earlier observations in Figure 5 and Figure 6.

In addition, the overall statistical analysis between vigilance group and enhancement group (taking all levels together) based on node strengths shows a significant difference between mental states as shown in Figure 13. This indicates that the audio stimulation significantly enhanced the mental state regardless of the levels of TOT. This supports our primary claim that audio can reduce vigilance decrement. From Figure 13, it is also obvious that the right hemisphere is highly sensitive to vigilance decrement. In this context, the group analysis supports the results obtained by individual level analysis (the analysis between the three levels of TOT).

#### 3.4.3. Clustering Coefficients and Characteristics Path Length

The average CC and PL measured under thresholding (T) between 0.1 ≤ T ≤ 0.3 is shown in Figure 14. In the brain functional network, the average clustering coefficient computed from all subjects is shown in Figure 14a. The average CC significantly decreased with increasing TOT in the vigilance state labelled with V1, V2, and V3, *p* < 0.05. Meanwhile, the average CC remains high with increasing TOT during the enhancement phase as labelled with E1, E2, and E3, respectively. The statistical analysis showed a significant increase in the clustering coefficients from vigilance to enhancement state at the three levels of TOT, *p* < 0.005. The increase in the CC in the enhancement phase indicates high local efficiency of information transfer.

Likewise, the characteristic path length result shows a significant increase with increasing time-on-task in the vigilance state as shown in Figure 14b, with mean *p* < 0.01. Correspondingly, the average PL remains small during the enhancement phase from the beginning of the experiment to the end of the experiment, 60 min. The statistical analysis showed that there was a significant decrease in the path length from vigilance to enhancement phase, *p* < 0.005. The decrease in path length from vigilance to enhancement indicates that the overall information processing ability is improved.

In addition, the average clustering coefficients and characteristic path lengths for the vigilance group and enhancement group show a significant difference between their characteristics as shown in Figure 15. This indicates that the audio stimulation significantly enhanced the mental state regardless the level of TOT. This supports our primary claim that audio reduced vigilance decrement.

#### 3.4.4. Hemispheric Analysis

Figure 16 shows the results of information transfer in hemispheric analysis. We measured the information transfer from the left hemisphere to the left hemisphere LH → LH, information flow from the right hemisphere to the right hemisphere RH → RH, information flow from the right hemisphere to the left hemisphere RH → LH, and the information flow from the left hemisphere to the right hemisphere LH → RH. The results in Figure 16a show that the information transfer from LH → LH and LH → RH are greater than the information transfer in other hemispheres for vigilance group at the three levels of time-on-task. The flow of information is more towards the left hemisphere from the left hemisphere nodes/electrodes and toward the right hemisphere from the left nodes. Thus, it can be said that, in vigilance state, the left region of the brain has received/sent more information from its own nodes than the right-side nodes. Additionally, the results show that the information transfer between hemispheres decreases with time on task from vigilance level 1 to vigilance level 2 and from vigilance level 2 to vigilance level 3. Note that the small circles in Figure 11a in the RH → RH and RH → LH violin graph are outliers.

Meanwhile, Figure 16b shows the information transfer for the enhancement group. The results show that the information transfers from LH → LH and RH → LH are greater than the information transfer from LH → RH at the three levels of time-on-task. The overall flow of information is more towards the left hemisphere from the right and left nodes/electrodes. Thus, it can be said that, in enhancement state, the left region of the brain has received more information from its own nodes as well as from the right-side nodes.

The statistical analysis of the differences in information transfer between vigilance levels and between vigilance and enhancement groups is given in Table 2. The results show that the differences in the mean information transfer are significant. The highest significant level was between vigilance and enhancement group while the least significant was in-between vigilance levels. The results support the hypothesis that audio stimulation can enhance the connectivity network and improve performance.

### 3.5. Relationships between Functional Connectivity Estimators and Behavioral Responses

In this section, we correlated the changes of the connectivity degree (nodal degree) with the reaction time (response time to stimuli) at the three levels of time-on-task. First, we measured the changes of the degree (Δ Degree) in each level between vigilance group and enhancement group by subtracting the degree of vigilance group from the degree of the enhancement group. The subtraction was based on one-versus-all operation. In this context, the averaged degree of each subject in the control group had the chance to be subtracted from that in all other subjects in the enhancement group. Then the final degree was based on weighted sum. Positive degree indicates enhancement, and negative degree indicates impairments or decrease in connectivity networks. Second, we measured the differences in the reaction time by subtraction the RT of vigilance group from that in the enhancement group. Positive reaction time indicates impairments, and negative time indicates enhancement. Third, we investigated the correlation between the changes on degrees of the functional connectivity at the three levels with the differences in the reaction time. The results show that the changes in degrees were negatively associated with changes in RT at the three levels of time-on-task as shown in Figure 17. Participants who showed the greatest increase in the degree of connectivity showed decrease in the reaction time to stimuli. This indicates that the higher the connectivity network between brain regions, the better the performance.

## 4. Discussion

In this study, we have experimentally examined the hypothesis that connectivity network patterns decrease with time on task while performing semantic complex vigilance task and are enhanced using audio stimulation. The vigilance state was induced using a complicated version of SCWT and enhanced by utilizing audio stimulation of pure tone at 250 Hz, presented simultaneously to the right and left ear of the participants. Vigilance enhancement was confirmed by the behavioral responses as well as the subjective evaluations. In general, the behavioral results confirmed that vigilance decrement was elicited by prolonged task performance; task performance became less efficient with time-on-task, response speed decreased, and the number of errors increased. The objective EEG results showed that the nodal degree, nodal strengths, and clustering coefficients of the connectivity network patterns were significantly reduced/weakened, and the characteristic path length was increased as time passed without audio stimulation. Meanwhile, the nodal degree, node strengths, and characteristic path length of the connectivity network patterns had significantly increased or maintained a high level when participants experimented while listening to the audio stimulation. The increase in the nodal degree, nodal strengths, and clustering coefficients during the enhancement phase was positively associated with task performance.

In this study, we found that audio stimulation significantly influenced the transient states of the mood of all participants, as well. Without audio stimulation, participants felt more angry, depressed, fatigued, confused, less calm, and unhappy after performing SCWT for 60 min. However, in the enhancement phase, all participants felt relatively calmer, happier, with more vigor, less depressed, and less fatigued when they performed the task while listening to audio stimulation. Thus, comparing the responses of vigilance decrement and enhancement groups showed that audio stimulation had a positive effect on the mood state and improved their performance as demonstrated by BRMUS questionnaires. Such results confirm that a shift in mood occurs due to the high cognitive load involved in performing SCWT, requiring sustained attention to the colors randomly presented. Besides, the result indicates that the proposed semantic vigilance task was resource-demanding and can be associated with a high cognitive workload.

The typical finding in vigilance decrement research is that cognitive efficiency declines over time due to the effect of mental fatigue. In line with that, our results of the reaction time (RT) support this claim. The increase in RT was also positively associated with the rise in the number of errors, omission and commission errors. The presented behavioral results, i.e., the increase in subjective anger, depression, fatigue, and confusion and the significant decrease in accuracy and increase in RT as a function of time-on-task are characteristic of a vigilance decrement [59,60,61]. Thus, cognitive efficiency decrement occurred because a subject expends mental resources for maintaining attention at a rate faster than they can be replenished. Mackworth et al. [62] first assessed that the accuracy of signal detections declined by 10% to 15% after about 30 min and continued to decrease after that gradually. Commensurate with these earlier findings, we found an increase followed by a decrease in response times within the first 15 min of the experiment and then a stabilized sharp rise to the end of the experiment. Meanwhile, in the enhancement phase, we have found that there was a decrement in the reaction time associated with improvements in the accuracy of detection, more specifically during the last 20 min. The effect of audio stimulation on the performance was prominently visible from the reaction time observed in the enhancement phase of the experiments. Furthermore, the decrease in the number of omission errors, as well as the decrement in commission errors also ascertains that audio stimulation enhanced vigilance.

The results of brain connectivity network analysis showed that vigilance decrement had weakened the nodal degree, clustering coefficients and nodal strengths as time passes. The decrease in the node degree, clustering coefficients and strengths of the network from the first block of time-on-task to the end of the experiment suggests that the ability to sustain attention decreased over time. More importantly, we found that both the parietal-to-frontal and parietal-to-temporal connectivity network decreased with time. It has been shown previously that frontal-to-parietal direction of information flow within EEG functional coupling is an intrinsic feature of brain network connectivity [63]. Similar to our results in Figure 5, Figure 6, Figure 7, Figure 10 and Figure 13, studies have found that when mental fatigue levels increase, the functional coupling decreases, specifically over the parietal-to-frontal regions in theta, alpha, and beta frequency bands [44,64,65]. It can be speculated that frontal-to-parietal EEG functional connectivity could reflect sensory signals from frontal to parietal areas. In this way, the frontal-to-parietal connectivity might be actively regulated by frontal-to-parietal signals, and their strengths might decrease with increasing time-on-task. This is consistent with the notion that the integration of information processed in the cerebral cortex might hinge on the formation and disappearance of synchronized neurons characterized by various frequency bands [65].

In addition, we also found that both occipital-to-frontal and occipital-to-temporal functional connectivity decreased with increasing time-on-task, as shown in Figure 5 and Figure 10. Meanwhile, it can be noted that the connected edges relatively concentrate in the left hemisphere during the first 40 min of vigilance and shifted to a bilateral style at the end of the experiment (last 20 min). The left hemisphere has previously been more closely linked to selective rather than sustained attention, showing activity in the color naming condition of a color/word stroop task and perceptual decision-making [66,67]. Our graphical annotation and statistical analysis of the results in Figure 6, Figure 7, Figure 8, Figure 9, Figure 10, Figure 11, Figure 12, Figure 13, Figure 14 and Figure 15 thus suggest that as vigilance increases, subjects cede control of selective attention. The flow of information sent/received to the right hemisphere was less than that in the left hemisphere over time. The disengagement of this region with vigilance could thus represent a weakening of the spatial engagement due to a decrease in motivation, which could further contribute to the reduction of reaction time. This finding is compatible with the results of Demeter et al. [68] showing that preserved right dorsolateral prefrontal function under conditions of fatigue is insufficient to sustain good performance. It is possible that under more difficult circumstances, right-hemisphere connectivity will be more severely weakened, leading to the more catastrophic lapses observed under these conditions. The reduction of connectivity on the right hemisphere in our study shown in Figure 9, Figure 10, Figure 13 and Figure 16 are also in line with previous studies that employed vigilance task of high complexity [69,70] and real-time driving [44,65].

Task difficulty has been shown to affect the laterality of vigilance processing. Researchers using transcranial Doppler sonography (TCD) have reported more significant declines in the right cerebral blood flow velocity (CBFV) from baseline values than the left CBFV during vigilance tasks [71,72]. Helton and his colleagues found that the easier the vigilance task was, the higher was the activation in the right hemisphere. Besides, they found less activation in the right hemisphere under difficult vigilance task [69]. It seems, therefore, unlikely that the hard task added processing load to the left hemisphere. Besides, previous research has indicated that traditional vigilance tasks are high in workload and are stressful [73]. Hence, the decrement in connectivity over the right hemisphere in our study is also in line with the previous stress studies that utilized EEG and functional near-infrared spectroscopy in their analysis [51,52,74,75,76]. Consequently, the shift in connectivity from unilateral to bilateral activation at the end of experiments indicates that increasing time-on-task induces a processing strategy change. It can be seen from the above overall analysis that, when task difficulty increased, the task increased demands on selective attention [77]. Thus, the results strongly support the position that increasing task difficulty induces a processing strategy change, from unilateral to more bilateral activation. More importantly, this finding is consistent with overload accounts, in which information processing resources are depleted over time, increasing task difficulty and leading to vigilance decrement.

During the vigilance enhancement phase, the results of connectivity in Figure 5b showed a similar complex network, linked by many edges between frontal, temporal, central, parietal, and occipital regions. The nodal degree, clustering coefficients and nodal strengths were maintained at high values from the first 20 min to the end of the experiment (small decrement in the node degree was found after 20 min but not significant). The higher connectivity between brain regions, while performing the vigilance task and listening to the audio stimulation, indicates that the overall information processing improved and may reflect a state of excitement. A possible aspect is that multisensory stimulation enhances sustained attention. One could argue that pure tone does not typically increase cognitive workload; rather, it activates peripheral regions of visual cortex (connections between the auditory cortex and visual cortex) and thus enhance performance [78,79,80]. Another interpretation of this result is that the higher connectivity with time-on-task was driven by the increased compensatory effort by participants to remain attentive for stimulus onset. In a related study, Wang et al. [81] used a 160 min cued stroop task to investigate compensatory brain activity in response to cognitive fatigue. They found that the amplitude of the anterior frontal ERP increased during their compensatory period, and the performance improved as time passed. Likewise, Matsubara et al. [82] utilized tone burst stimuli of 500 Hz presented monaurally to the patient with ipsilateral mesial temporal lobe epilepsy. They found it to evoke more neural synchronization in the auditory cortices. Meanwhile, Helton et al. [83] reported better performance in high salience of noise than low salience. The study found that noise elevated task, and low signal salience elevated distress. Similar to our results, higher connectivity has also been observed in the context of music perception when compared to silence [84]. One may conclude that higher nodal values across the cortex facilitate neural communication, promote neural plasticity, and enhance vigilance, although, in older subjects, increased neural activity was found to be related to performance improvement [85].

It should be noted, however, that the small decrement in the node degree from the first 20 min to the last 20 min as shown in Figure 8b might be due to habituation. Nevertheless, habituation is defined as the behavioral response decrement that results from repeated stimulation and does not involve response fatigue. Interestingly, we did not find any decrement in the behavioral responses. One of the possibilities is the hormone release [86] which is the final output of the neuroendocrine system; hormones have a persistent action in regulating many behaviors. Rausch and his colleagues [87] suggested that white noise enhances phasic dopamine release, thereby modulating activity within the superior temporal sulcus and leading to increased attention and memory formation. Another reason might be the involvement of multisensory process (integration of visual and auditory information) [88]. In this context, habituation may require longer time and 60 min may not show significant habituation.

In the comparison of connectivity network analysis between the vigilance phase and enhancement phase at the three levels of time-on-task, the largest increase was observed in the nodal degree and nodal strengths between the occipital-to-frontal, frontal-to-parietal, and frontal-to-temporal (see the topographical maps in Figure 10 and Figure 13). Interestingly, more improvement in connectivity network patterns was found in the right hemisphere. The information flow from the right hemisphere to left hemisphere and within right hemisphere itself significantly increased with audio stimulation. This indicates that audio stimulation caused the right hemisphere to engage in mental stimulation and increased motivation. These findings suggest that audio stimulation may be well-suited to mitigate performance degradation in work settings requiring sustained attention. Our results fit well within the resource theory of vigilance and suggest that resource utilization may be augmented via exogenous stimulation. The overall results of the present study indicate that the increase in brain connectivity network is associated with better performance, reduced reaction time, and improved overall accuracy on the vigilance task.

### Limitations of the Study

This study investigated the impact of audio stimulation on vigilance decrement with time-on-task. The effects after stimulation on performance are yet to be explored, whereby more studies having a large sample size are needed to support further the technique’s effectiveness for reducing vigilance decrement. Second, although EEG signal is one of the most predictive indicators and works well in mental state discrimination, it may correlate less with behavior to some extent. Studying vigilance using multi-modal neuroimaging, including functional near-infrared spectroscopy (fNIRS) and functional magnetic resonance imaging (fMRI) simultaneously with the EEG would be an advantage. This will aid in understanding the neural mechanisms underlying the vigilant state. Third, although the statistical analysis between the two mental states based on directed comparison is fair enough to show the differences, multiple comparison correction should be considered in future studies. Fourth, a continuous prolonged audio stimulus of more than 60 min should be conducted in future studies to see whether subjects become habituated to the tone.

## 5. Conclusions

In summary, the behavioral responses in the vigilance state showed that the speed of reaction time to stimuli and accuracy decreased with increasing time-on-task (TOT). On the contrary, we observed an increase in accuracy and speed of reaction to stimuli with increasing time-on-task, with audio stimulation. Similarly, the subjective evaluations showed that the factors of fatigue, depression, and anger significantly decrease while happy and calmness factors increased in the enhancement phase compared to vigilance phase. Correspondingly, brain connectivity network analysis in the vigilance state showed decrement over time and was maintained high or enhanced with audio stimulation. During the vigilance state, we found that the connected edges relatively concentrated in the left hemisphere during the first 40 min and shifted to bilateral style at the end of the experiment (last 20 min). Besides, the vigilance connectivity network analysis showed weakened parietal-to-frontal, parietal-to-temporal, occipital-to-frontal, and occipital-to-temporal network that is consistent with the well-mapped network of sustained attention. Meanwhile, audio stimulation maintained high/increased degree of connectivity network over the entire brain regions from the beginning to the end of the experiment. The increase in the degree of connectivity network correlates with individual differences in the magnitude of the vigilance enhancement assessed by response time to stimuli. The results provide evidence for the improvement of cognitive processing efficiency brought on by audio stimulation, leading to enhanced processing and increase of target detection.

## Figures and Tables

**Figure 1 brainsci-09-00363-f001:**
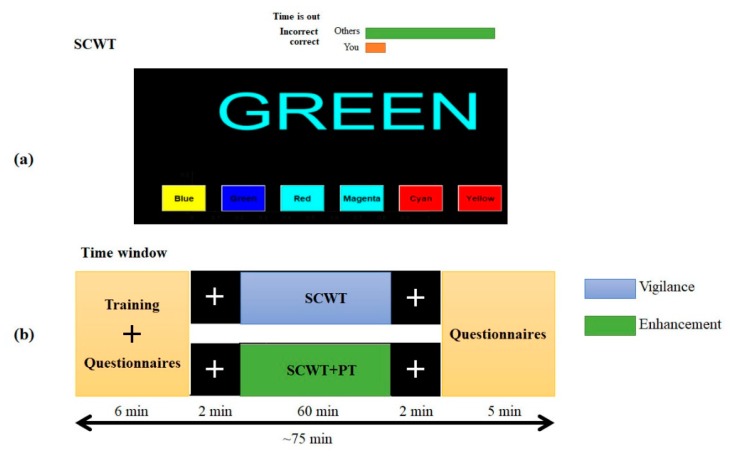
Experimental protocol (**a**) stroop color-word task (SCWT) presentation interface and (**b**) timing window. In the timing window, the plus sign in black background is for the pre- and post-baseline. Sixty (60) min SCWT is for the vigilance group, and the 60 min stroop color-word task with pure tone, SCWT+PT, is for the enhancement group.

**Figure 2 brainsci-09-00363-f002:**
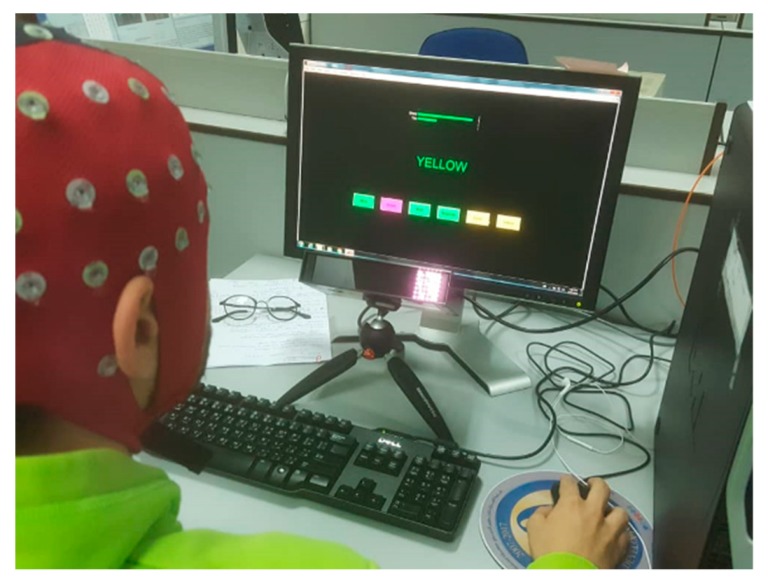
EEG data acquisition and experimental set-up.

**Figure 3 brainsci-09-00363-f003:**
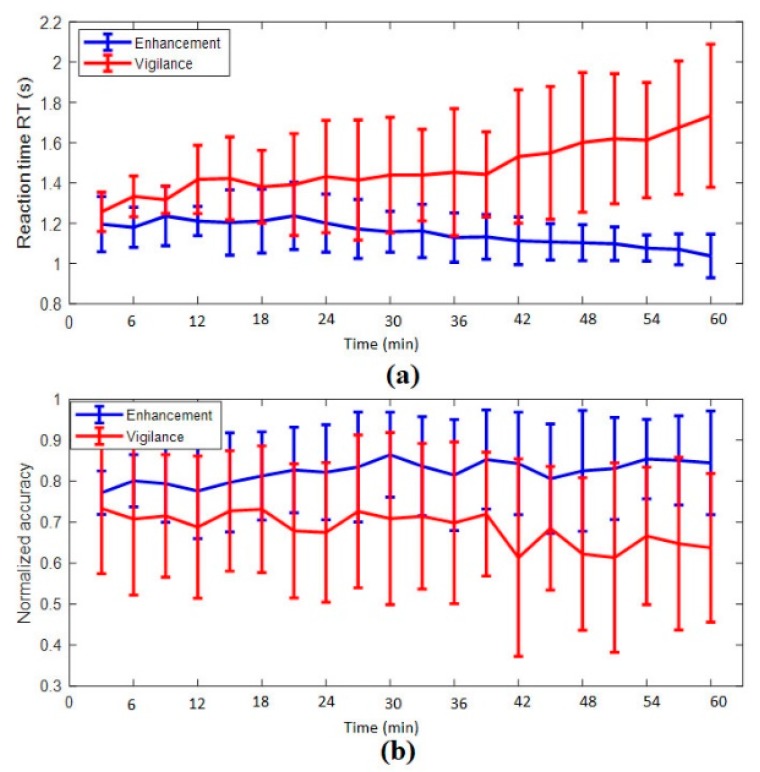
Performance evaluation with time on task (**a**) reaction time and (**b**) accuracy score. Each time interval is averaged at every 3 min.

**Figure 4 brainsci-09-00363-f004:**
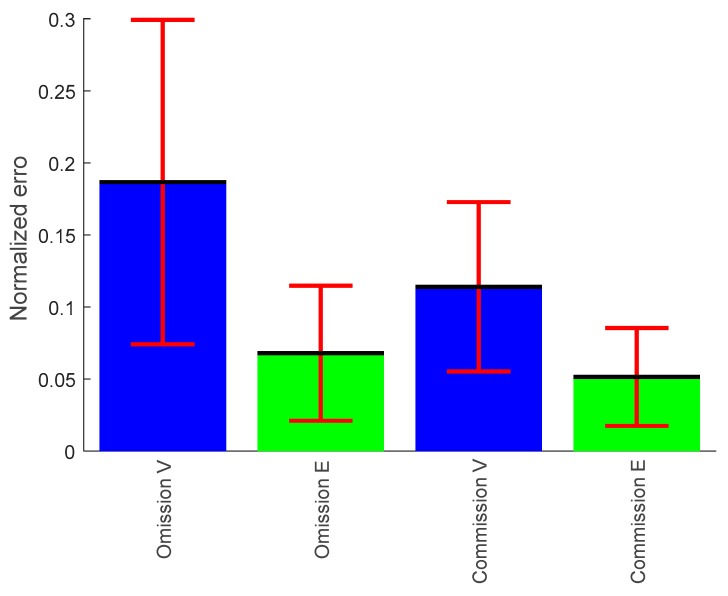
Percentage of omission and commission error in the vigilance (V) and enhancement state group (E).

**Figure 5 brainsci-09-00363-f005:**
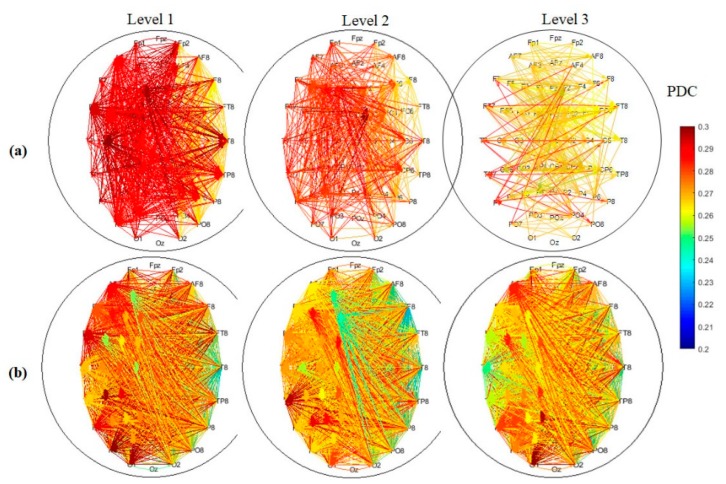
Average weighted directed connectivity network for (**a**) vigilance levels, (**b**) enhancement levels. The connectivity network at Level 1 is measured at time between 0 to 20 min, meanwhile the connectivity network at Level 2 is measured at time between 21 to 40 min, and the connectivity network at Level 3 is measured between 41 to 60 min, respectively. The color bar represents the PDC thresholding values ≥0.2. Red indicates high connectivity strength and blue indicates less connectivity strength.

**Figure 6 brainsci-09-00363-f006:**
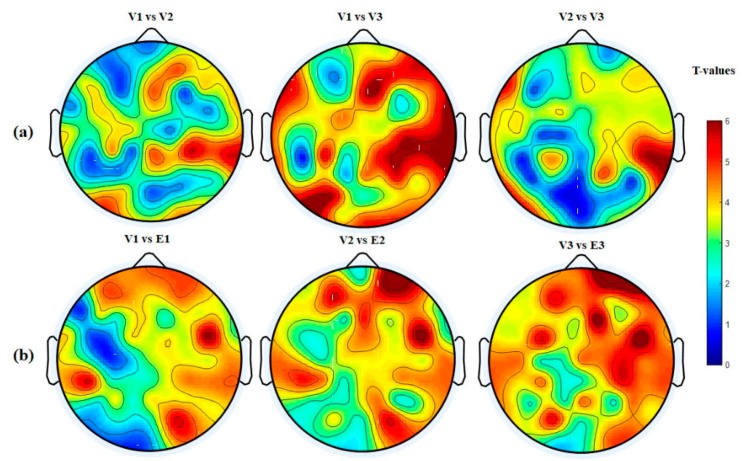
Statistical analysis between node degree indexes of connectivity network at three levels of time-on-task. (**a**) Vigilance group and (**b**) vigilance versus enhancement levels. The variables V1, E1 represent vigilance and enhancement at Level 1; V2, E2 represent vigilance and enhancement at Level 2; and V3, E3 represent vigilance and enhancement at Level 3. Red color indicates highly while blue color indicates less significant differences between brain connectivity networks.

**Figure 7 brainsci-09-00363-f007:**
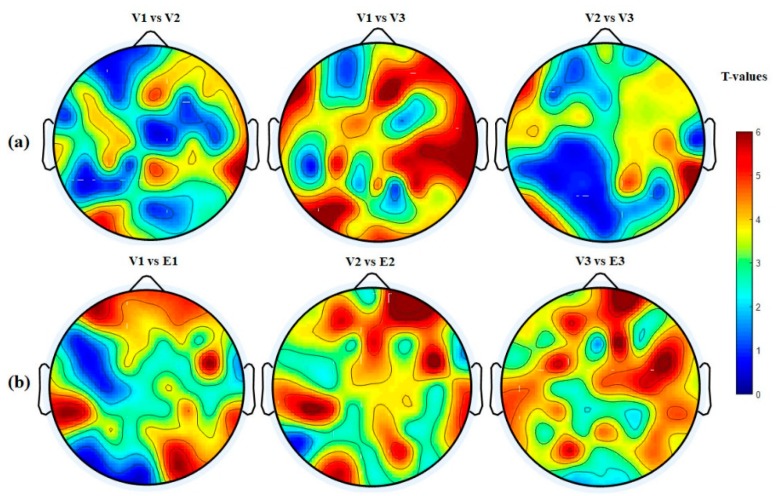
Statistical *t*-test topographical maps of EEG electrodes between (**a**) vigilance levels and (**b**) vigilance versus enhancement levels. The variables V1, E1 represent vigilance and enhancement at Level 1; V2, E2 represent vigilance and enhancement at Level 2; and V3, E3 represent vigilance and enhancement at Level 3. Red color indicates highly significant while blue color indicates less significant differences between brain connectivity networks.

**Figure 8 brainsci-09-00363-f008:**
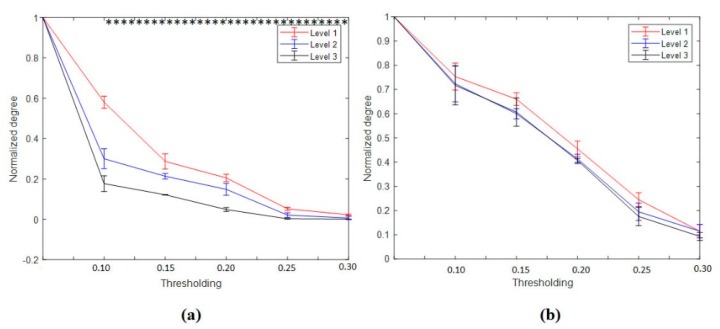
Average global degree index of connectivity network at three levels of time-on-task. (**a**) Vigilance group and (**b**) enhancement group. The bars represent mean ± standard deviation between subjects and the * shows that the differences between vigilance levels is significant at given threshold value, *p* < 0.01.

**Figure 9 brainsci-09-00363-f009:**
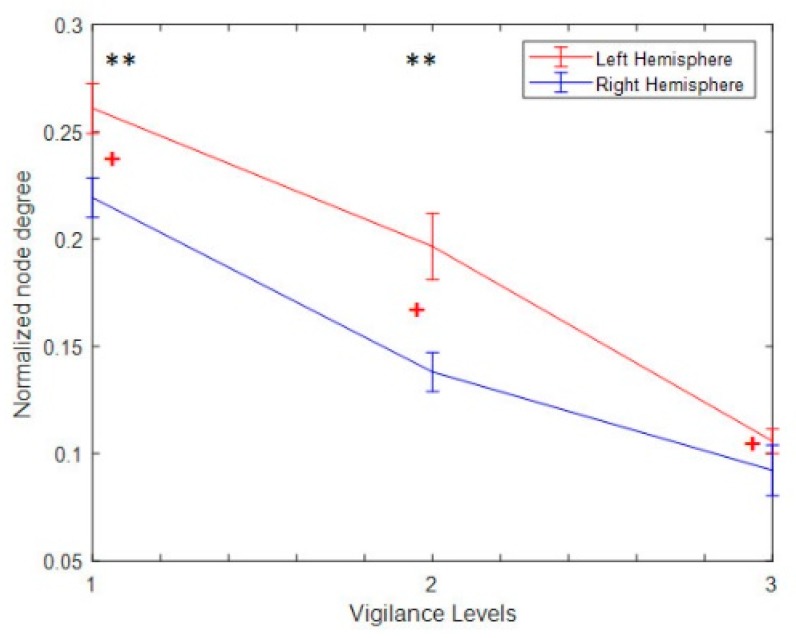
Normalized node degree of vigilance at the three levels of TOT. The red line represents the average node degree on the left hemisphere, and the blue line represents the average node degree of the right hemisphere. The black ** indicated the differences between the left and right hemisphere is significant, and the red ^+^ indicates the normalized nodal degree on the left hemisphere is greater than the right hemisphere based on lateralized index calculated using LH-RH/LH+RH. The RH and LH stand for left and right hemisphere respectively.

**Figure 10 brainsci-09-00363-f010:**
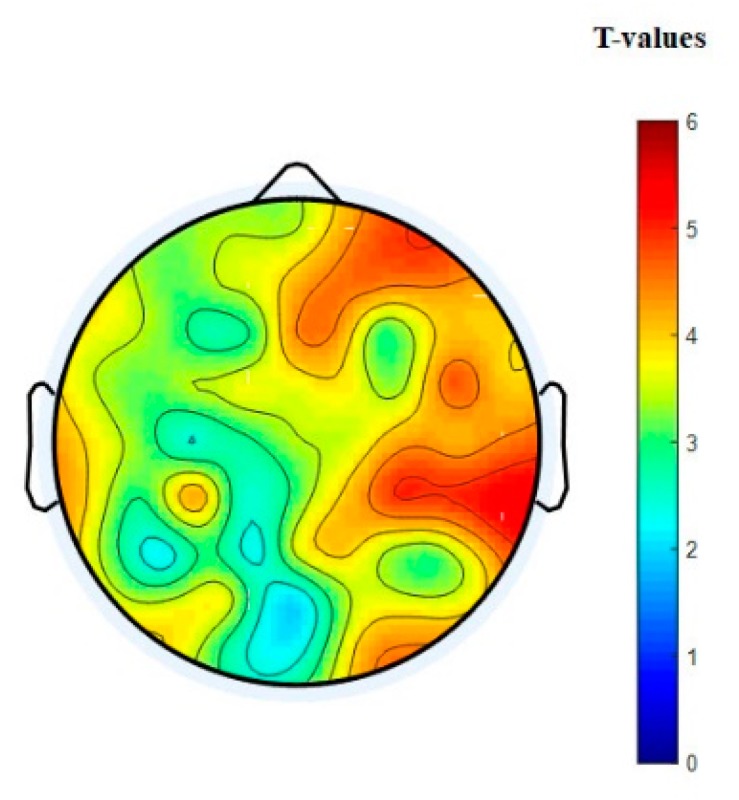
Statistical *t*-test topographical map of EEG electrodes between the three levels of vigilance mental state and the three levels of enhanced mental state based on node strengths.

**Figure 11 brainsci-09-00363-f011:**
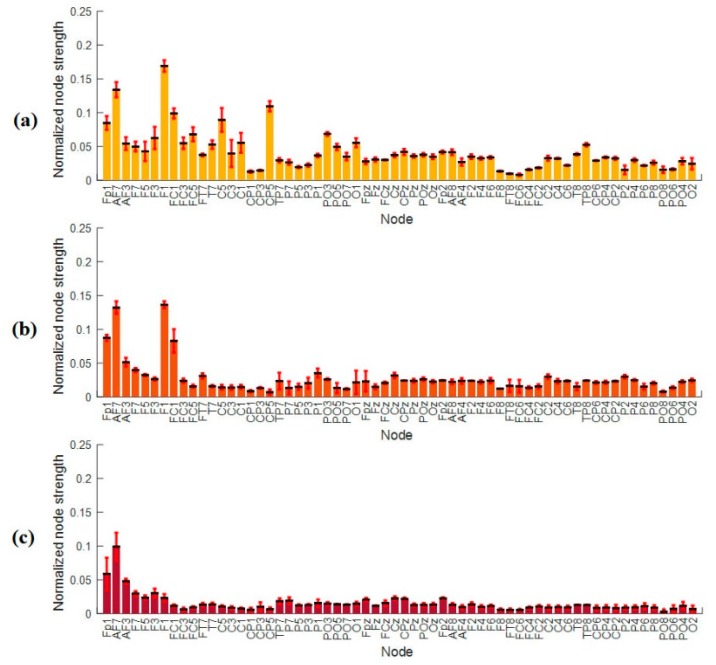
Average node strength across electrodes during (**a**) vigilance level 1, (**b**) vigilance level 2, and (**c**) vigilance level 3. The bars represent mean ± standard deviation between subjects at individual electrode.

**Figure 12 brainsci-09-00363-f012:**
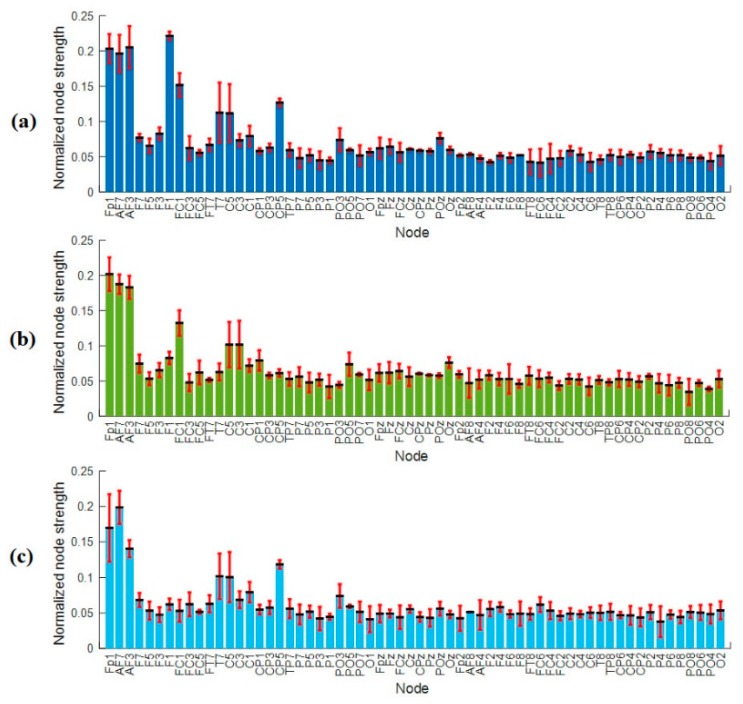
Average node strength across electrodes during (**a**) enhancement level 1, (**b**) enhancement level 2, and (**c**) enhancement level 3. The bars represent mean ± standard deviation between subjects at individual electrode.

**Figure 13 brainsci-09-00363-f013:**
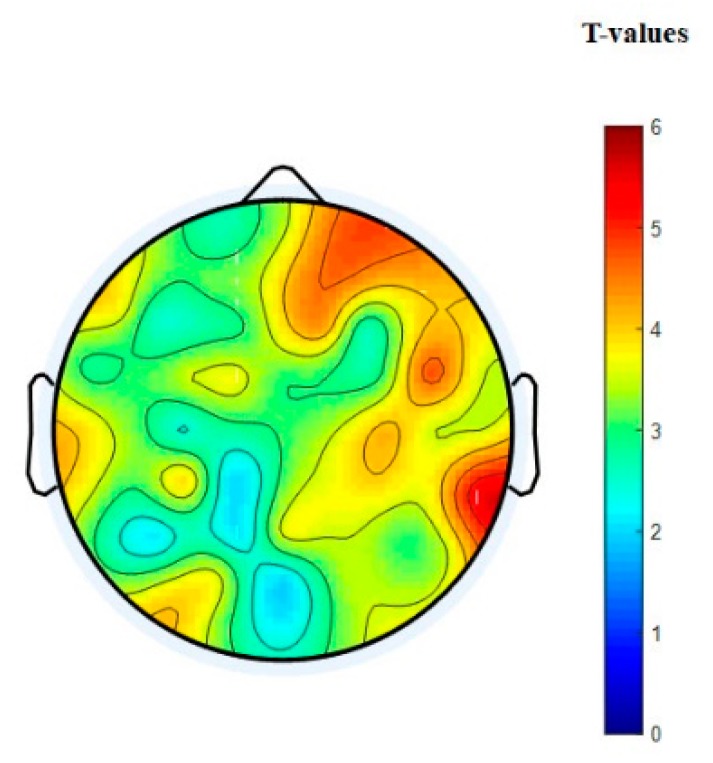
Statistical *t*-test topographical map of EEG electrodes between the three levels of vigilance mental state and the three levels of enhanced mental state based on node strengths.

**Figure 14 brainsci-09-00363-f014:**
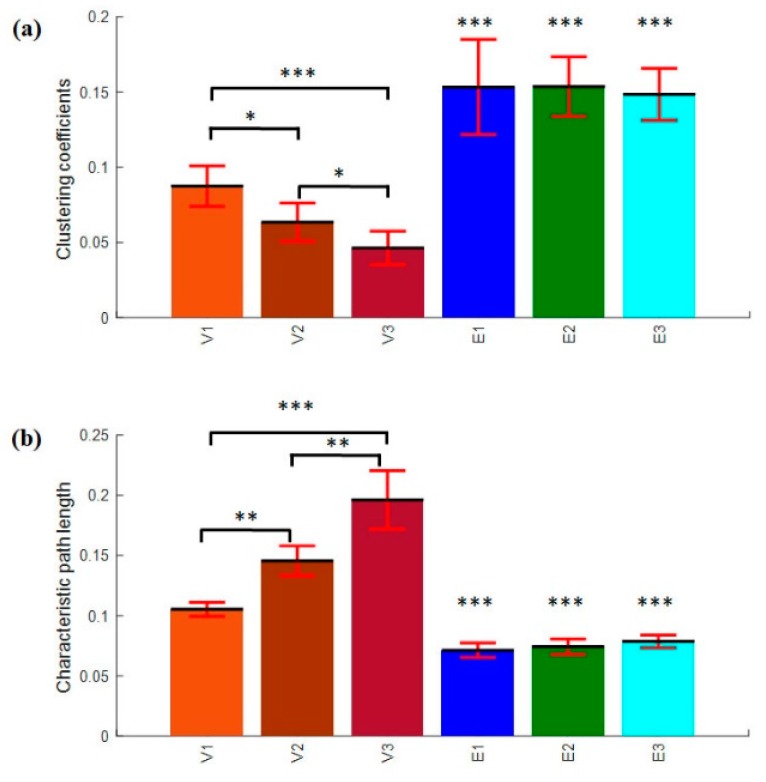
(**a**) Clustering coefficients and (**b**) characteristic path length of the vigilance and enhancement states. The variables V1, V2, and V3 represent vigilance state at Level 1 of time-on-task, Level 2, and Level 3. Likewise, the variables E1, E2, and E3 represent enhancement state at Level 1 of time-on-task, Level 2, and Level 3. The error bars represent the standard deviation across subjects. The marks ‘*’, ‘**’ and ‘***’ indicate that the differences between the two conditions/levels is significant with *p* < 0.05, *p* < 0.01 and *p* < 0.001, respectively.

**Figure 15 brainsci-09-00363-f015:**
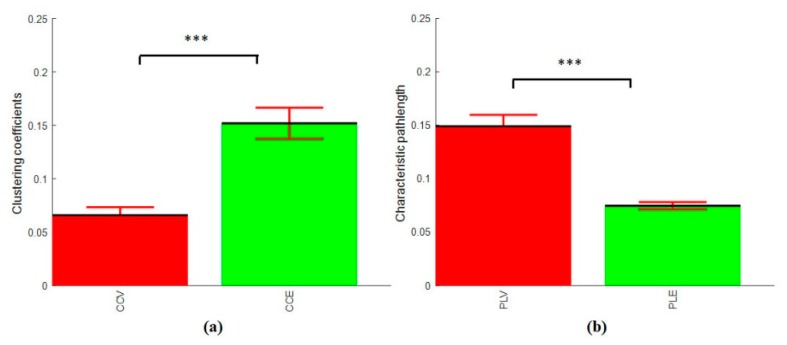
(**a**) Average clustering coefficients of the three levels of TOT. The CCV stands for the clustering coefficients at vigilance decrement mental state, and CCE stands for the clustering coefficients at the enhanced mental state. (**b**) Average characteristic path length of the three levels of TOT. The PLV stands for the characteristic path length at vigilance decrement mental state, and PLE stands for the characteristic path length at the enhanced mental state. The error bars represent the standard deviation across subjects. The marks ‘***’ indicate that the differences between the two conditions is significant with *p* < 0.001.

**Figure 16 brainsci-09-00363-f016:**
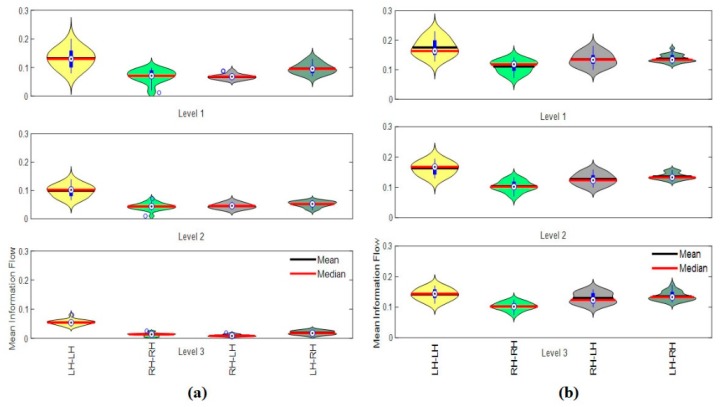
Hemispheric mean information flow at three levels during (**a**) vigilance and (**b**) enhancement. The acronym LH stands for left hemisphere, and RH stands for the right hemisphere.

**Figure 17 brainsci-09-00363-f017:**
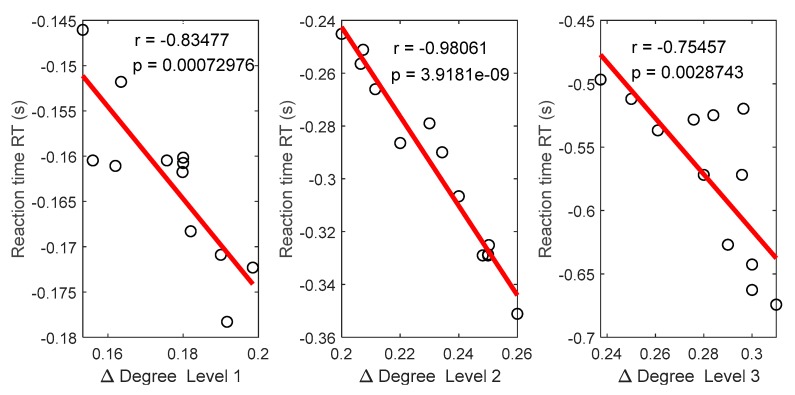
Scatter plots showing the correlation between the subjects’ reaction time and the nodal degree in Level 1, Level 2, and Level 3, respectively. The horizontal axis represents the values of the respective graph metrics, and the vertical axis stands for the reaction time. The *r*- and *p*-values of the corresponding correlations are displayed in the figures.

**Table 1 brainsci-09-00363-t001:** Average BRMUS scores for 32 items after 60-min vigilance task with no tone and after 60-min enhancement with 250-Hz pure tone; SD indicates the standard deviation of the difference between subjects.

Factors	Items	Vigilance Mean ± SD	Enhancement Mean ± SD	Significance
**Anger**	Annoyed	1.50 ± 1.51	0.90 ± 1.22	↓↓
	Bitter	0.83 ± 1.16	0.27 ± 0.46	↓↓
	Angry	0.50 ± 1.22	0.27 ± 0.90	↓
	Bad tempered	0.66 ± 1.21	0.54 ± 0.93	
**Tension**	Panicky	0.66 ± 1.21	1.10 ± 1.59	↑↑
	Anxious	0.66 ± 1.21	1.09 ± 1.51	
	Worried	1.16 ± 1.16	1.0 ± 1.54	
	Nervous	0.16 ± 0.40	0.63 ± 0.28	
**Depression**	Depressed	0.50 ± 0.54	0.27 ± 0.46	↓
	Downhearted	0.66 ± 1.21	0.36 ± 0.67	↓↓
	Unhappy	0.83 ± 1.16	0.27 ± 0.46	↓↓
	Miserable	1.16 ± 1.47	0.36 ± 0.92	↓↓
**Vigor**	Lively	2.00 ± 1.41	2.27 ± 1.42	
	Energetic	1.66 ± 1.36	2.0 ± 1.48	↑↑
	Active	2.16 ± 1.83	2.27 ± 1.81	
	Alert	2.16 ± 1.83	2.27 ± 1.71	
**Fatigue**	Worn out	2.16 ± 1.32	1.18 ± 1.32	↓↓
	Exhausted	2.33 ± 1.50	1.36 ± 1.50	↓↓
	Sleepy	2.33 ± 1.36	1.36 ± 1.36	↓↓
	Tired	2.83 ± 1.16	1.72 ± 1.48	↓↓
**Confusion**	Confused	0.66 ± 0.40	0.30 ± 0.30	↓
	Muddled	0.66 ± 1.03	0.18 ± 1.40	↓
	Mixed up	0.66 ± 0.81	0.60 ± 1.30	
	Uncertain	1.00 ± 0.89	0.54 ± 0.93	↓
**Happy**	Cheerful	2.00 ± 1.26	2.63 ± 1.40	↑↑
	Content	1.66 ± 1.21	2.27 ± 1.27	↑↑
	Happy	2.33 ± 1.21	2.27 ± 1.19	
	Satisfied	1.66 ± 1.12	2.45 ± 1.36	↑↑
**Calmness**	Calm	2.16 ± 1.47	2.45 ± 1.03	↑↑
	Composed	2.00 ± 0.63	2.45 ± 0.93	↑↑
	Relaxed	1.83 ± 1.16	2.45 ± 1.21	↑↑
	Restful	1.36 ± 1.36	2.00 ± 1.34	↑↑

The up arrows indicate that the differences significantly increase and the down arrows indicate the differences significantly decrease. A single indicator is corresponding to *p* < 0.05.

**Table 2 brainsci-09-00363-t002:** Statistical analysis of hemispheric information flow in-between vigilance levels and between vigilance and enhancement levels.

Hemisphere	V1 vs. V2 *p*-Value (<)	V1 vs. V3 *p*-Value (<)	V2 vs. V3 *p*-Value (<)	V1 vs. E1 *p*-Value (<)	V2 vs. E2 *p*-Value (<)	V3 vs. E3 *p*-Value (<)
**LH → LH**	0.0411	0.0002	0.0401	0.0201	0.0041	0.0001
**RH → RH**	0.0001	0.0001	0.0051	0.0021	0.0001	0.0001
**RH → LH**	0.0401	0.0011	0.0031	0.0001	0.0001	0.0001
**LH → RH**	0.0311	0.0001	0.0031	0.005	0.0051	0.0001

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
