# Peer review of "Brain Connectivity Analysis Under Semantic Vigilance and Enhanced Mental States"

_brainsci, 2019, doi:10.3390/brainsci9120363_

Round 1

Reviewer 1 Report

The paper describes an experiment to examine how an auditory stimulus can enhance vigilance and performance in a standard stroop-color task. Furthermore, EEG is used to access connectivity based on graph theoretical analysis.
Major comments:
1. The authors mention in the introduction that the majority of studies that analyse responses over time on time does not involve traditional vigilance paradigms. This doesn't seem correct. The authors should do a more careful review in particular of the effect of auditory stimulation on vigilance.
2. Statistical analysis: Statistical analysis should involve multiple comparison correction. This applies to figure 9, figure 10 and table 2. At the moment, no concrete conclusions can be derived.
3. Figures 7-8 would be better if they are represented with topographic brain maps.
4. It's unclear why in figure 3 the reaction time and accuracy is observed up to 20 minutes, whereas the overal experiment lasts 60 minutes.
5. The authors should discuss whether subjects could get habituated to a continous long auditory stimulus and when.

Author Response

The authors would like to thank the Reviewer for sparing the time and effort in reviewing our manuscript and for the positive recommendation and insightful and helpful comments. The comments directed our attention to important areas that need improvement leading to a stronger paper. In this response (attached file), we explain how each of the comments is addressed. Please note that the reviewer’ comments are in black, while our response and modified text are in blue.

Reviewer 2 Report

Dear Authors, 

First of all, I sincerely apologize my review process is really delayed.  

In the present paper, using a combination of EEG recording, functional connectivity analysis (PDC) and graph theory analysis, the authors attempted to reveal what neural basis is behind the vigilance. I totally agree with the importance of the authors' idea. In addition, the method to record and preprocess the EEG data has no problem.  However, unfortunately, I found that most of the authors' claims seem not to be supported by the data shown here. The authors have to add some analyses to support their claims.   

Major points

 (L363-364) Authors claim that ‘large quantities of edges between parietal and temporal regions gradually vanish’ based on Figures 5. However, who on the earth could understand what the authors want to claim based on this messy graph? Authors have to add some analyses that could make potential readers easily understand the vanishment of connectivity involving the parietal and temporal area. For example, take an average of connectivity involving the TP8 in each graph, then we could get 6 (3 by 2) summarized values about average weighted connectivity strength. It is possible to do a statistical analysis of the summarized data. If you could find a decrease of the summarized weighted connectivity strength in the TP8 but not in O1, we could easily understand the connectivity vanishes specifically in the temporal/parietal region, as the authors claimed in the manuscript. Regarding this issue, what electrodes have to be categorized as electrodes covering temporal and parietal regions? The authors have to add this information.

(L364-365) The claim in this paragraph, left-right asymmetry and time dependency of connectivity strength is also not supported by the data. When I see Figure 5a, I could have the impression that connectivity involving electrodes located on the left hemisphere is greater than the opposite.  However, this is just a visual impression.  Authors must add some analysis that could show hemispheric asymmetry and time dependency of connectivity decrease over time.

(Figure 6) Regarding this figure, I have many comments.  First, the authors said that they did the statistical analysis to show the difference between the vigilance and enhancement group. In the paragraph from L293 (2.5. Statistical analysis), the pairwise t-test was used to compare the connectivity metrics between the vigilance and enhancement. I think it is not a conventional way to handle this data type. The first choice should be the repeated measures ANOVA, and if you found that the variance between three conditions (V1, V2 and V3) is not identical then you have to move to the post-hoc analysis. I know some statistician says the ANOVA process is not necessary, and we could directly start from the direct comparison between three groups (i.e., V1 vs V2, V2 vs V3, V3 vs V1). I’m not a statistician, but I could agree with this idea. However, even in the case, the correction for multiple comparisons is indispensable.  There is no description in this paper. The Bonferroni is too conservative, but at least the B-H method should be used, to avoid the possibility that what you found is a false-positive. I write this comment about statistics here, but this comment is not limited to the results based on Figure 5. I strongly recommend authors re-check whether their statistical analysis is appropriate for, and also I strongly recommend authors to add more detailed information about the statistics according to the APA guideline (i.e., https://www.statisticshowto.datasciencecentral.com/reporting-statistics-apa-style/). It could improve what you did in your statistical analysis. Second, why the authors summarized (averaged) the nodal degree index of all electrodes, while the authors claimed there is a hemispheric and regional asymmetry of connectivity? What I mean is authors claim relating to Figure 5. Averaging degree involving all electrodes misses important information. Authors have to discuss the hemispheric/regional asymmetry of degree by adding analyses or justify averaging all electrodes while electrodes on left and right hemisphere has different property. The following points are relatively minor comments but regarding the preparing this Figure 6, why authors selected the specific thresholding values of >0.10? Is this arbitrary selection, isn’t it? In addition, while the X-axis (horizontal-axis) is named ‘Thresholding T’, what is this?

              (Figures 7 and 8) What I could say to your result is almost same.  The discussion written in the manuscript is not supported by rigid data analysis, but just standing on visual impression.  For example, from L403, the authors claimed that ‘we can observe that, vigilance group has more node strength within the left hemisphere compared to the right hemisphere’. The electrodes with odd numbers seem to have high normalized nodal strength, but that is just ‘looks like’. Please add information and statistical analysis, that could make us easy to understand the authors claim is standing on the data authors acquired. In addition, as I repeatedly pointed out, one of the important findings in this manuscript is vigilance might have an effect specifically on the parietal/temporal area. Therefore, even in this node strength analysis, I recommend authors to consider this regional inhomogeneity. It could enhance the quality and impact of the paper.  What I would like to say relating to Figure 8 is almost identical as mentioned above: the conclusion should not be based on visual inspection/impression, authors need to do a statistical comparison and make clear why authors selected an arbitrary threshold.  I know thresholding is a very hard problem, and there is no clear consensus. However, you should add a description of why the authors believed the value is appropriate.

              (Figure 9) Why do t>2.5 means indicate that the differences were significant? It might be related to the degree of freedom and sample size, I guess. The authors should add a description why thresholding value t=2.5 is reasonable.

(Figure 10) What kind of clustering coefficient was calculated in your paper? Global CC? Local CC? In addition, if my understanding is correct, there is a way to calculate the CC based on directed connectivity. This paper’s connectivity is originally calculated by the PDC, this is directed connectivity measure.  Then, have authors used the directed (and weighted) CC? Need more detailed description of what the authors did to calculate it. Without this, I have no chance to judge whether what the authors reported in Figure 10 is correct or not.

(Figure 12) Number of data points, indicated by the black circle, seems to be different across three levels: I see 13 for Level1, but 12 for Levels 2/3. What is 13? In my understanding, 24 participants were recruited, and they were divided into two sub-groups: the vigilance group (maybe, 12 participants), and the enhancement group, right? If my understanding is correct, one subject did either ‘vigilance experiment’ or ‘enhancement experiment’. Then, how authors subtracted the ‘changes of the degree (L495)’. It’s possible that I’m doing a misunderstanding, but please explain this in more detail.

Minor comments

Why authors averaged the PDC values to depict the graph? The PDC is directed, but by averaging it, it loses the information of information flow. It loses the strong point of the directed coherence, and then, simply calculating Pearson’s correlation is enough to do the same thing.

(Figure 11) What is a small circle? Outlier?

Author Response

(The authors gave the same response as above.)

Round 2

Reviewer 1 Report

The paper has improved significantly with the latest revision and results have been clarified. The authors have addressed most of my comments. In terms of statistical analysis the authors have not used Anova but they incorporated more rigorous clarification and they focus their results based on the comparison of two conditions.

Minor comment: Figure3: Best to keep axis in seconds, although they clarify this in the caption, it creates confusion.

Author Response

The authors would like to thank the Reviewer  for sparing the time and effort in reviewing our manuscript and for the positive recommendation and insightful and helpful comments. The comments directed our attention to important areas that need improvement leading to a stronger paper. In this response, we explain how each of the comments is addressed. Please note that the reviewers’ comments are in black, while our response and modified text are in blue.

Please see the responses to the comments attached in the word file. 
